# Cost–Utility Analysis of Tenofovir Alafenamide and Entecavir in Chronic Hepatitis B Patients: A Markov Decision Model

**DOI:** 10.3390/cancers16040813

**Published:** 2024-02-17

**Authors:** Chun-Huang Lai, Hon-Yi Shi, Cheng-En Tsai, Yuan-Chieh Yang, Si-Un Frank Chiu

**Affiliations:** 1Division of Gastroenterology, Department of Internal Medicine, Kaohsiung Municipal United Hospital, Kaohsiung 80457, Taiwan; lch21422@kmuh.gov.tw (C.-H.L.); cetsai0213@kmuh.gov.tw (C.-E.T.); 2Department of Healthcare Administration and Medical Informatics, Kaohsiung Medical University, Kaohsiung 80708, Taiwan; 3Department of Business Management, National Sun Yat-Sen University, Kaohsiung 80424, Taiwan; 4Department of Medical Research, Kaohsiung Medical University Hospital, Kaohsiung 80708, Taiwan; 5Department of Medical Research, China Medical University Hospital, China Medical University, Taichung 40433, Taiwan; 6Department of Laboratory Medicine, Kaohsiung Municipal United Hospital, Kaohsiung 80457, Taiwan; yang1967@kmuh.gov.tw; 7Department of Computer Science, Brown University, Providence, RI 02912, USA; si-un_chiu@brown.edu; 8Department of Economics, Brown University, Providence, RI 02912, USA

**Keywords:** chronic hepatitis B, tenofovir alafenamide, entecavir, inverse probability of treatment weighting, cost–utility analysis

## Abstract

**Simple Summary:**

This study conducted a cost–utility analysis comparing tenofovir alafenamide (TAF) and entecavir (ETV) as first-line treatments for chronic hepatitis B. The TAF group exhibited an incremental cost effectiveness ratio (ICER) of −NT$23,878 per quality-adjusted life year (QALY) compared to the ETV group. Additionally, TAF demonstrated superior cost effectiveness, suggesting potential annual savings of over NT$500 million (approximately US$18 million). These findings support the consideration for expanding health insurance coverage for TAF in hepatitis B treatment.

**Abstract:**

From the perspective of health economics, the evaluation of drug-related cost effectiveness and clinical utility is crucial. We conducted a cost–utility analysis of two first-line drugs, tenofovir alafenamide (TAF) and entecavir (ETV), in the treatment of chronic hepatitis B (CHB) patients. We performed inverse probability of treatment weighting (IPTW) to match the independent variables between the two treatment groups. The incremental cost effectiveness ratio (ICER) of the two treatment groups was simulated using a decision tree with the Markov annual-cycle model. A total of 54 patients treated with TAF and 98 with ETV from January 2016 to December 2020 were enrolled. The total medical cost in the TAF group was NT$76,098 less than that in the ETV group, and TAF demonstrated more effectiveness than ETV by 3.19 quality-adjusted life years (QALYs). When the time horizon was set at 30 years, the ICER of the TAF group compared with the ETV group was −NT$23,878 per QALY, suggesting more cost savings for TAF. Additionally, with the application of TAF, over NT$366 million (approximately US$12 million) can be saved annually. TAF demonstrates cheaper medical costs and more favorable clinical QALYs than ETV. To balance health insurance benefits and cost effectiveness, TAF is the optimal treatment for CHB.

## 1. Introduction

Chronic hepatitis B (CHB) is a significant global health concern, with over 290 million carriers worldwide, leading to more than 820,000 deaths annually and exceeding US$2 billion in associated costs [1]. In Taiwan, where over 10,000 individuals succumb to cirrhosis or hepatocellular carcinoma each year, half of these cases are attributed to chronic hepatitis B, indicating substantial health risks and economic burdens [2].

The current pharmacological approaches for treating CHB involve long-acting interferon and oral antiviral drugs [3,4]. Interferon, administered through injections, presents challenges due to suboptimal rates of e antigen (HBeAg) seroconversion and surface antigen (HBsAg) seroclearance, coupled with notable adverse effects [5]. Oral antiviral drugs, such as tenofovir disoproxil (TDF), tenofovir alafenamide (TAF), and entecavir (ETV), offer more sustained options. However, the long-term use of TDF is associated with potential renal and skeletal issues, with TAF emerging as an alternative with a more favorable side effect profile [6,7,8,9,10].

Although oral antiviral agents demonstrate efficacy in suppressing the hepatitis B virus (HBV), the attainment of clinical significance typically requires a minimum of three years. Additionally, CHB is characterized by a substantial relapse rate upon the discontinuation of antiviral medications, requiring repeated treatment cycles. The cessation of TAF therapy has demonstrated significantly earlier and higher clinical relapse rates compared to ETV therapy [9,10]. This may necessitate lifelong medication for some patients, resulting in substantial medical expenditures [11]. From a health economics perspective, the assessment of drug-related cost effectiveness and clinical utility assumes paramount importance. A limited number of studies have undertaken an exploration of the health economics associated with oral antiviral drugs in the treatment of CHB, and scant attention has been devoted to a comparative analysis of the cost–utility relationship between TAF and ETV [6,7,8]. Moreover, prior comparative investigations involving TAF and ETV have predominantly employed simulated patient groups rather than individuals treated in real-world clinical settings.

Building upon these research backgrounds and motivations, we undertook a pharmacoeconomic analysis to scrutinize the treatment costs and clinical efficacy of TAF and ETV in real-world CHB patients. The primary aim was to identify the optimal first-line oral antiviral drug for practical implementation in clinical settings.

## 2. Materials and Methods

### 2.1. Research Samples and Study Design

This retrospective study enrolled CHB patients treated with TAF or ETV (covered by the National Health Insurance), who visited a regional teaching hospital or a district teaching hospital in southern Taiwan from January 2016 to December 2020. The investigators reviewed medical records to obtain the patients’ demographic data and disease characteristics. The health insurance claims were also reviewed to collect the medical costs for each patient after a one-year treatment of TAF or ETV. The exclusion criteria included patients under 18 years old or with any malignant tumors. A total of 54 patients (TAF group) and 98 patients (ETV group) were enrolled in this study (Figure 1). Before the study’s initiation, approval was obtained from the Institutional Review Board of our hospital (KSPH-2022-06).

### 2.2. Confounding Variables

By retrospectively reviewing the medical records of the patients, we identified confounding variables associated with the demographic and clinical characteristics, including age, sex, hepatitis B e antigen (HBeAg) (positive or negative), and viral loads before treatment.

### 2.3. Economic Evaluation

This was a payer’s perspective pharmacoeconomic study, and data involving the direct medical cost were retrieved from the health insurance claims of the National Health Insurance Research Database (NHIRD). The total direct medical costs denote the one-year outpatient and inpatient expenses per patient upon initiating either TAF or ETV treatment. Specifically, outpatient costs consisted of fees for physicians, laboratory and image diagnosis, medicines, and pharmacists, while inpatient costs included fees for physicians, radiology image diagnosis, pharmacists, nursing care, accommodation charges, laboratory testing, and medicines. 

### 2.4. Cost–Utility Analysis

We used the cost–utility analysis to conduct the decision analysis and to investigate whether the first-line drug TAF is more cost effective than ETV for CHB patients. The Markov model utilized in this study was developed by integrating insights from local clinical practices and referencing pertinent studies, as illustrated in Figure 2 [3,4,5,6]. This model categorizes post-treatment conditions of patients receiving the first-line drug into four distinct states: CHB, hepatocellular carcinoma (HCC), cirrhosis, and mortality. According to local clinical experience, we set one cycle at one year and the research time horizon at 30 years. It is assumed that all the patients with hepatitis B were initially in a survival and responding state; then, their original state is set based on the evaluation results. Afterward, patients’ categories may vary depending on their updated states in each cycle. The state could be transferred directly to the death stage, which remains the same, permanently, without further transfers.

### 2.5. Statistical Analysis 

The unit of the analysis in this study was the individual patient with CHB. Continuous variables, including age, medical costs, and viral loads, were described using the mean and standard deviation. Categorical variables, including sex and HBeAg, were described using the frequency and percentages. 

Inverse probability of treatment weighting (IPTW) was employed for variable matching [12], and a decision tree with the Markov model was used for dynamic simulations to conduct the cost–utility analysis of the TAF and ETV. We used IPTW to match the four variables (age, sex, HBeAg, and viral load) before the intervention. The weight calculation was as follows: exposed group (TAF group) = 1/propensity score; unexposed group (ETV group) = 1/(1 − propensity score). In the present study, the differences between individual variables in the two groups were described using standardized differences, and 10% was considered as a significant difference.

The EQ-5D-3L scores extracted from the literature review were transformed to utility values utilizing the time tradeoff (TTO) formula by incorporating the Taiwan coefficient [13,14,15]. Within the TTO valuation process, respondents initially assessed whether a given health state was superior to, equivalent to, or inferior to death. In instances where a state was perceived as superior to death, respondents were subsequently queried about the number of years (*t*) at which they would be indifferent between *t* years of life in optimal health and 10 years of life in that specific state. The TTO value for that particular state was computed as *t*/10 (0 < *t* ≤ 10). For states considered as unfavorable as death, the assigned TTO value was 0. In situations where a state was regarded as worse than death, patients were prompted to specify t at which they would be indifferent between a life of (10 − *t*) years in that state followed by *t* years in full health and death. The TTO value for states worse than death was then determined as −*t*/(10 − *t*). Each cost structure was adjusted to the present value in 2021 according to Taiwan’s consumer price index (CPI), and the cost and quality-adjusted life years (QALYs) in the decision-making model were discounted at a rate of 2%. In addition, for the cost–utility analysis, the between-group difference in the total direct medical costs and QALYs was expressed as the incremental cost effectiveness ratio (ICER). The cost–utility in the TAF group compared with that in the ETV group was used as the baseline. The ICER equals the incremental cost (i.e., the total direct medical cost in the TAF group minus the total direct medical cost in the ETV group) divided by the marginal QALYs (i.e., the QALYs in the TAF group minus the QALYs in the ETV group).

In addition to various costs, requisite parameters and non-parameters for the Markov decision-tree model in this study included the clinical transfer probabilities between states and the utility values corresponding to each state [14,15,16]. These values were meticulously curated from pertinent research, with probabilities, costs, and utilities [17,18,19,20,21]. The optimal path selection for the decision analysis was scrutinized employing the rollback method, a technique facilitating the retrospective tracing of total costs and treatment outcomes resulting from the implementation of distinct strategies.

The study devised a Markov decision-tree model with predetermined parameters encompassing utility values, transfer rates, costs, and benefits for each state. Simulating patient state transitions during the intervention and aggregating outcomes informed the identification of the more cost effective alternative through Markov decision-tree analysis. The probabilistic sensitivity analysis (PSA) comprised 1000 Monte Carlo simulations, yielding ICER values. The graphical representation employed cost–utility acceptability curves and scatter plots. A one-way sensitivity analysis, manipulating costs and outcomes by ±20%, discerned influential variables presented through tornado diagrams. Additionally, the willingness to pay (WTP) was set at the gross domestic product (GDP) per capita in 2022, namely, NT$990,120.

We also used SPSS v. 23.0 (SPSS, Chicago, IL, USA) for the descriptive and inferential statistics and used TreeAge Pro 2021 version (Tree-Age Software, Williamstown, MA, USA) for the cost–utility analysis. 

## 3. Results

Before the IPTW, the average age in the TAF group and the ETV group was 52.9 ± 11.2 and 60.3 ± 13.8 years old, respectively (Table 1). The between-group standardized difference was −58.8%, suggesting a significant difference. As for sex, the TAF group enrolled 75.9% men and 24.1% women, while the ETV group included 75.5% men and 24.1% women. The between-group standardized difference was 0.93%, indicating no significant difference. After IPTW matching, the standardized differences of the demographics were reduced to less than 10%, suggesting no statistical difference between the TAF and ETV groups.

The health insurance claims of the patients one year after initiating antiviral drugs for hepatitis B were divided into outpatient costs, inpatient costs, and total one-year medical costs. After IPTW matching, the mean outpatient costs per patient in the TAF and ETV groups were NT$60,241 ± 8033 and NT$62,956 ± 13,867, respectively (Table 2). The mean inpatient costs per patient in the TAF and ETV groups were NT$80,868 ± 33,999 and NT$87,387 ± 101,745, respectively. The mean total direct medical cost from both the outpatient and in-patient costs within 1 year was NT$141,109 ± 120,398 for the TAF group and NT$150,343 ± 152,825 for the ETV group.

The articulated Markov decision-tree model is visually represented in Figure 3. The time horizon of the Markov decision-tree model simulation was delineated over a 30-year trajectory. Within this temporal framework, the total costs for individuals within the TAF group were registered at NT$1,241,822, corresponding to an effectiveness of 14.63 QALYs. In contrast, the ETV group incurred total costs of NT$1,317,920, accompanied by an effectiveness of 11.44 QALYs (Table 3). Significantly, the TAF strategy exhibited dominance over the ETV strategy, manifesting heightened cost effectiveness and resulting in demonstrable cost savings.

As presented in Figure 4, as the WTP increases, the net monetary benefit (NMB) in the TAF group was utterly better than that in the ETV group. Additionally, the CUAC revealed that when the WTP was between NT$0 and NT$990,120, the probability for the TAF group being more cost effective than the ETV group was 100% (Figure 5). 

The scatter plot of the PSA obtained from the incremental cost–utility analysis revealed that at a WTP of NT$990,120, all the parameters were centered in the fourth quadrant, indicating that 100% of the TAF group was more cost effective compared with the ETV group (Figure 6). Figure 7 shows the tornado diagram of the one-way sensitivity analysis. The parameters were analyzed individually, and the parameters contributing to ICER value are listed in their order of significance: the cost of the TAF for HCC, followed by the cost of the TAF for hepatitis B, the cost of the ETV for hepatitis B, the cost of the ETV for cirrhosis, and the cost of the TAF for cirrhosis.

## 4. Discussion

This study, following IPTW matching and a simulation spanning a 30-year time horizon, revealed that the TAF group displayed reduced total costs and enhanced QALYs compared to the ETV group. These findings suggest that TAF proved to be a more cost-saving option than ETV. 

The National Health Insurance Plan in Taiwan initiated the coverages of ETV in 2016 and TAF in 2019. As a part of routine procedures, negotiations between the National Health Insurance Administration and pharmaceutical companies occur annually to establish prescription drug prices. The initial cost of brand-name ETV was NT$152 per pill, progressively reducing to NT$130 per pill in 2019, aligning it with the cost of TAF. Following adjustments based on the annual consumer price index (CPI), the costs of ETV and TAF became comparable. In contrast to international studies reporting TAF being 18.5%–56.4% more expensive than ETV in cost–utility analyses [22,23], the pharmaceutical cost of TAF in Taiwan is notably economical. Consequently, within the Taiwanese context, TAF presents the optimal cost–utility profile.

This study boasts several methodological strengths: (a) IPTW for variable matching: A salient feature is the application of IPTW for meticulous variable matching. Notably, substantial differences in age, HBeAg, and viral loads between the intervention groups, with standardized differences surpassing 10%, were effectively rectified through IPTW matching. This methodological choice not only serves to mitigate potential research bias but also ensures the preservation of small, yet valuable, samples [24]. (b) Real-world medical costs for analysis: A distinctive facet of this study lies in the utilization of real-world medical costs for the analysis. By acknowledging the potential variability in outcomes from simulated cost–utility analyses of TAF for hepatitis B patients in different clinical scenarios [22,23], this study’s reliance on real-world data provides a more authentic basis for evaluating the disparities in the actual costs and utility between TAF and ETV. (c) One-on-one comparison of cost–utility profiles: This study represents a pioneering effort in conducting a direct comparison of the cost–utility profiles of TAF and ETV. This methodological approach enhances the precision and granularity in the assessment, yielding valuable insights into the nuanced cost effectiveness and utility of these interventions.

In clinical settings, TAF has demonstrated superiority over alternative oral medications, showcasing effective viral response inhibition, viral load suppression, liver enzyme regulation, resistance prevention, and kidney function preservation [10,25]. Despite its recent market introduction, TAF’s effects are well documented [8,9,10]. A rigorous study across ten U.S. medical centers, focusing on hepatitis B patients undergoing at least two years of TAF treatment, revealed significant improvements in achieving HBV DNA ≤20 IU/mL and reduced abnormal liver functions [26]. In contrast, a two-year course of ETV in the same study showed no significant differences in viral loads or liver functions. Notably, ETV-treated patients exhibited a higher risk of severe liver fibrosis compared to TAF-treated counterparts. A crossover study demonstrated a substantial reduction in HBsAg in patients transitioning from ETV to TAF [27]. In acute liver failure in-patients, TAF outperformed ETV in preserving kidney functions [28]. A three-year retrospective study found that TAF-treated patients experienced less significant kidney function deterioration compared to ETV-treated patients [29]. The majority of the studies focusing on the cost–utility analysis of antiviral hepatitis B drugs have primarily centered on TDF and ETV [30,31,32]. Given the relatively recent introduction of TAF to the market, only a limited number of studies have explored its cost–utility profile [22,23]. Notably, two studies employed Markov simulations to evaluate TAF, TDF, and ETV, both concluding that TDF was the most cost effective treatment. However, owing to TDF’s documented adverse effects on kidney functions and bone-mass density [9,10], TAF is emerging as an alternative to TDF in clinical practice [26]. Consequently, a critical gap exists, necessitating a one-on-one comparative study between TAF and ETV [33].

This study explores the economic burdens of hepatitis B in Taiwan, considering a 0.9% annual increase in carriers, of which approximately 19% are eligible for medication [34]. Transitioning from ETV to TAF as the first-line drug could yield annual savings exceeding NT$366 million (approximately US$12 million). This estimation is based on the difference in the total direct medical costs, population estimates, and the proportion eligible for treatment, as represented by the following formula: ((NT$150,343 − NT$141,109) × 23,196,178 people × 0.9% × 19% = NT$366,270,898) [35]. Globally, considering a 1.5 million annual increase in hepatitis B carriers, adopting TAF instead of ETV could result in substantial annual savings surpassing NT$2.6 billion (approximately US$88 million). This projection considers the same cost differential, global population estimates, and the proportion eligible for treatment, as expressed by the following formula: ((NT$150,343 − NT$141,109) × 1,500,000 people × 19% = NT$2,631,690,000 = US$87,723,000). This study presents certain limitations. First, the patient cohort is derived solely from one regional and one district hospital, omitting data from medical centers and potentially limiting the representation of patient types and intervention outcomes. Second, the study’s focus on TAF, a relatively recent addition to National Health Insurance coverage for only the last two years, resulted in a relatively small sample size. Consequently, stratification based on decompensated cirrhosis severity and the inclusion of comorbidity data were not feasible. Future investigations, encompassing a larger and more diverse patient population over an extended duration, are warranted for a comprehensive assessment of TAF’s effects on cirrhosis and comorbidities, providing more precise medical-cost and clinical-effectiveness data. Third, reliance on literature reviews for transition probabilities and utility values highlights a need for future studies incorporating real-world data for a more accurate understanding of hepatitis B transitions and the clinical utilities of TAF and ETV.

## 5. Conclusions

Tenofovir alafenamide (TAF) emerges as a more economically efficient intervention than entecavir (ETV) in the management of patients with chronic hepatitis B (CHB). This observation underscores TAF’s fiscal feasibility as the preferred choice for enhancing patients’ well-being, offering guidance to hospital administrators, and informing the National Health Insurance Administration. This study posits a compelling proposition for the augmentation of health insurance coverage to encompass TAF within the spectrum of CHB treatment modalities.

## Figures and Tables

**Figure 1 cancers-16-00813-f001:**
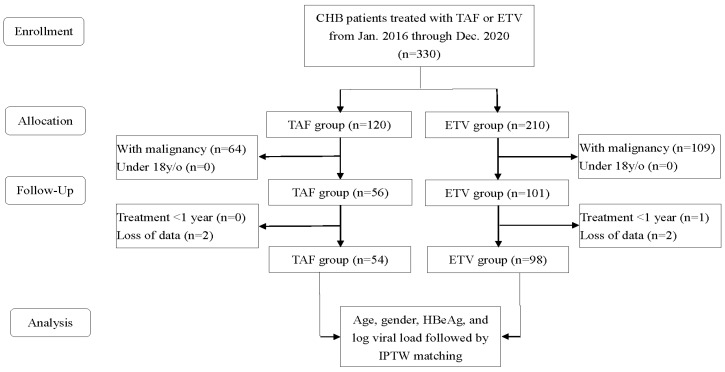
Flowchart of the study. CHB, chronic hepatitis B; TAF, tenofovir alafenamide; ETV, entecavir; IPTW, inverse probability of treatment weighting.

**Figure 2 cancers-16-00813-f002:**
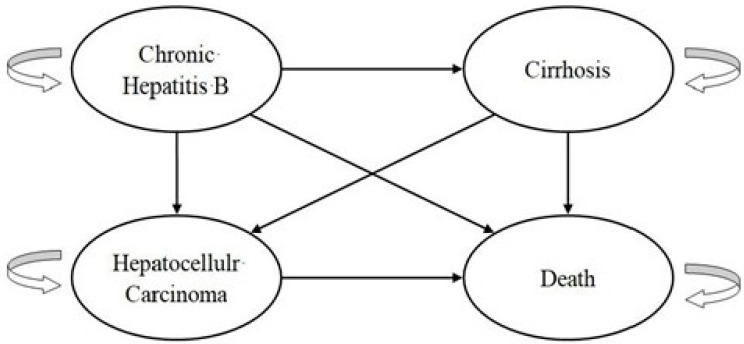
Schematic overview of the Markov model.

**Figure 3 cancers-16-00813-f003:**
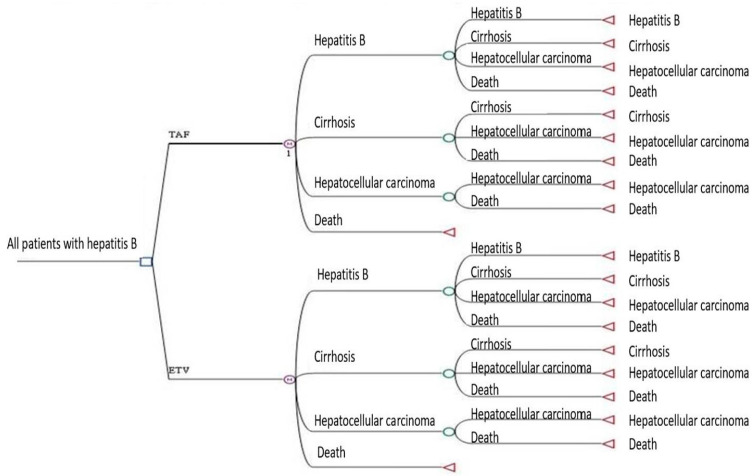
Markov decision-tree model for cost–utility analysis of two first-line drugs, tenofovir alafenamide (TAF) and entecavir (ETV), in the treatment of chronic hepatitis B (CHB) patients. M, Markov model; green cycle, choose code.

**Figure 4 cancers-16-00813-f004:**
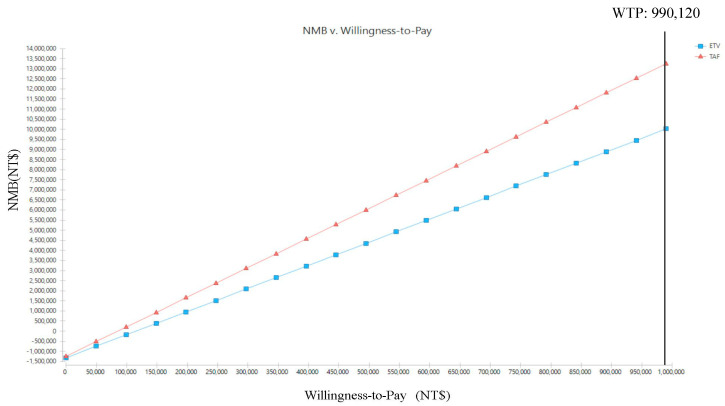
Net monetary benefit (NMB) of two first-line drugs, tenofovir alafenamide (TAF) and entecavir (ETV), in the treatment of chronic hepatitis B (CHB) patients.

**Figure 5 cancers-16-00813-f005:**
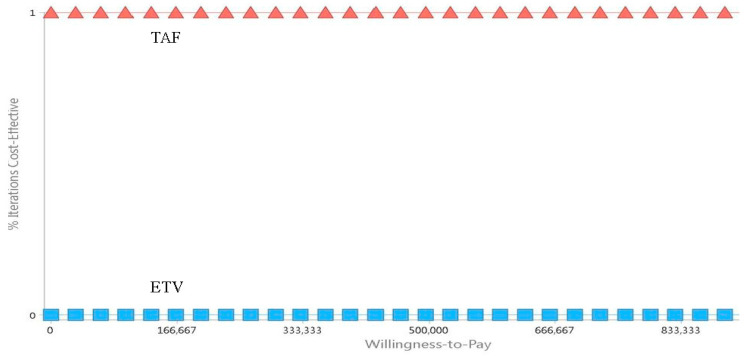
Cost effectiveness acceptability curves of two first-line drugs, tenofovir alafenamide (TAF) and entecavir (ETV), in the treatment of chronic hepatitis B (CHB) patients. Triangle means TAF, square means ETV.

**Figure 6 cancers-16-00813-f006:**
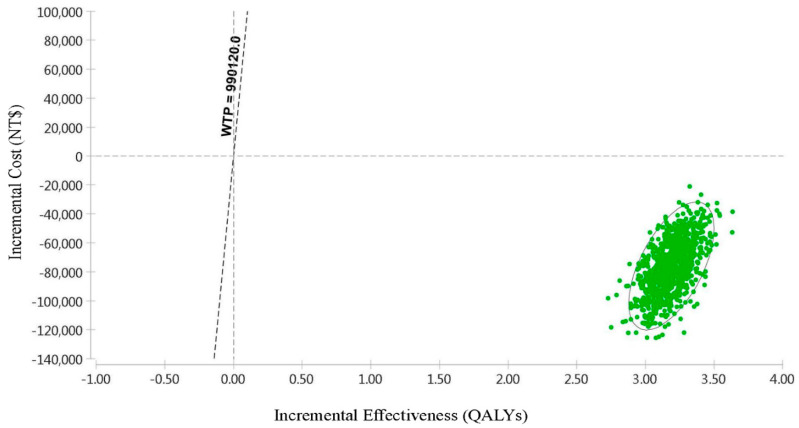
Incremental cost-effectiveness scatter plot of tenofovir alafenamide (TAF) versus entecavir (ETV) in the treatment of chronic hepatitis B (CHB) patients. WTP, willingness to pay; QALYs, quality-adjusted life years.

**Figure 7 cancers-16-00813-f007:**
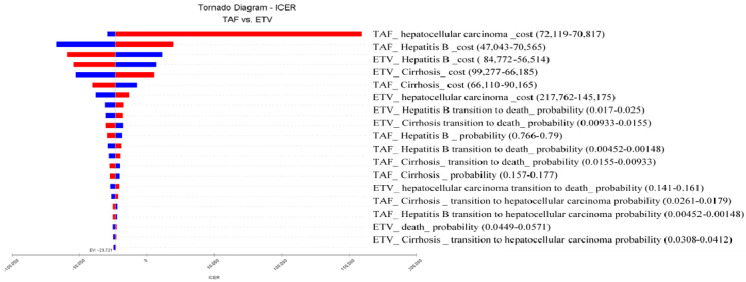
ICERs of TAF compared with ETV. Tornado diagram showing one-way sensitivity analysis results. Bars indicate the effect of a ±20% variance in a variable on the ICER. Costs are expressed in 2021 NT$. ICER, incremental cost effectiveness ratio; TAF, tenofovir alafenamide; ETV, entecavir. Additionally, blue bars indicate a positive effect, while red bars indicate a negative effect.

**Table 1 cancers-16-00813-t001:** Study characteristics before and after IPTW matching.

Variable		Total (*n* = 152) Mean ± Standard Deviation or Median (IQR)	Before IPTW			After IPTW		
			TAF (*n* = 54)	ETV (*n* = 98)	Standardized Difference (%)	TAF	ETV	Standardized Difference (%)
Age		57.66 ± 13.29 57.5 [46–67.75]	52.94 ± 11.02 50.0 [44.75–62.25]	60.27 ± 13.76 61 [47.75–72.0]	−58.78	56.52 ± 11.35 56.0 [46.0–66.0]	57.45 ± 13.78 58.0 [45.0–67.63]	−7.37
Gender	Male	115 (75.7%)	41 (75.9%)	74 (75.5%)	0.932	71.8%	74.3%	−5.62
	Female	37 (24.3)	13 (24.1%)	24 (24.5%)		28.2%	25.7%	
HBeAg	Positive	50 (32.9%)	26 (48.1%)	24 (24.5%)	50.62	33.6%	32.9%	1.49
	Negative	102 (67.1%)	28 (51.9%)	74(75.5%)		66.4%	67.1%	
Log Viral Load		5.97 ± 1.85 6.12 [4.53–7.81]	6.27 ± 1.85 6.80 [4.97–7.89]	5.81 ± 1.84 5.97 [4.24–7.63]	25	5.89 ± 2.00 6.38 [4.45–7.80]	5.96 ± 1.83 6.11 [4.30–7.66]	−3.65

IPTW, inverse probability of treatment weighting; IQR, interquartile range; TAF, tenofovir alafenamide; ETV, entecavir.

**Table 2 cancers-16-00813-t002:** Markov health-state transition probabilities, cost structure, and utility of TAF and ETV treatments.

	TAF	ETV	Distribution	Reference
**Transition probability**				
Chronic hepatitis B				
Chronic hepatitis B	96.58%	94.38%	Beta	[16]
Cirrhosis	2.82%	2.82%	Beta	[16]
Hepatocellular carcinoma	0.30%	0.70%	Beta	[16]
Death	0.30%	2.10%	Beta	[16]
Cirrhosis				
Cirrhosis	96.56%	95.16%	Beta	[17]
Hepatocellular carcinoma	2.20%	3.60%	Beta	[1]
Death	1.24%	1.24%	Beta	[17]
Hepatocellular carcinoma				
Hepatocellular carcinoma	84.90%	84.90%	Beta	[18]
Death	15.10%	15.10%	Beta	[18]
**Cost (NT$)**				
Outpatient (per cycle for 1 year)				
Medicine	50,912 ± 6826	52,143 ± 12,492	Gamma	This study
Physician	5374 ± 4540	6653 ± 4322	Gamma	This study
Laboratory	2928 ± 460	3098 ± 724	Gamma	This study
Pharmacist	1008 ± 122	990 ± 198	Gamma	This study
Total outpatient costs	60,241 ± 8033	62,956 ± 13,867		
Hospitalization (per cycle for 1 year)				
Medicine	10,730 ± 7181	12,876 ± 20,532	Gamma	This study
Physician	20,609 ± 11,598	15,037 ± 11,156	Gamma	This study
Laboratory	12,262 ± 4952	14,460 ± 13,337	Gamma	This study
Ward	25,645 ± 17,678	30,160 ± 32,093	Gamma	This study
Radiology	3748 ± 4069	7037 ± 14,222	Gamma	This study
Therapy treatment	7874 ± 4788	4954 ± 8452	Gamma	This study
Total hospitalization costs	80,868 ± 33,999	87,387 ± 101,745		
Total direct medical costs	141,109 ± 120,398	150,343 ± 152,825		
**Utility**				
Chronic hepatitis B	0.85 (0.68–0.89)	0.85 (0.68–0.89)	Beta	[15]
Compensated cirrhosis	0.69 (0.66–0.71)	0.69 (0.66–0.71)	Beta	[2]
Decompensated cirrhosis	0.35 (0.32–0.37)	0.35 (0.32–0.37)	Beta	[2]
Hepatocellular carcinoma	0.38 (0.36–0.41)	0.38 (0.36–0.41)	Beta	[2]
Liver transplantation	0.57 (0.54–0.60)	0.57 (0.54–0.60)	Beta	[2]

TAF, tenofovir alafenamide; ETV, entecavir; NT$, New Taiwan Dollars.

**Table 3 cancers-16-00813-t003:** Summary of Markov decision-tree model simulation analysis.

Treatment	Cost (NT$)	Incremental Cost (NT$)	QALYs	Incremental QALYs	NMB (NT$)	CE	ICER (NT$/QALY)
TAF	1,241,822	−76,098	14.63	3.19	13,241,786	84,893	−23,878
ETV	1,317,920		11.44		10,010,058	1,151,926	

TAF, tenofovir alafenamide; ETV, entecavir; NT$, New Taiwan Dollars; QALYs, quality-adjusted life years; NMB, net monetary benefit; CE, cost effectiveness; ICER, incremental cost effectiveness ratio.

## Data Availability

Data and study materials can be made available for non-commercial use upon reasonable request to the corresponding author.

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
