# Peer review of "Cost–Utility Analysis of Tenofovir Alafenamide and Entecavir in Chronic Hepatitis B Patients: A Markov Decision Model"

_cancers, 2024, doi:10.3390/cancers16040813_

Round 1
Reviewer 1 Report
Comments and Suggestions for Authors
General comment
The authors presented the cost-utility analysis of using TAF versus ETV in CHB patients based on real-world data, i.e. CHB patients treated with TAF or ETV (covered by the National Health Insurance), who visited a regional teaching hospital or a district teaching hospital in southern Taiwan from January 2016 through December 2020. Their main objective was to identify the optimal first-line oral antiviral drug in CHB patients. Data were collected retrospectively from medical records and health insurance claims: 54 patients in TAF group and 98 patients in ETV group. The analysis concludes that TAF is more cost-effective than ETV for treating CHB patients.
My general comment is that this conclusion was highly expected since progression is always faster and costs higher with ETV compared to TAF.
My specific comments could help ensure the methodology applied and the robustness of the conclusion.
Specific comments
-1. Introduction: “A limited number of studies have undertaken an exploration of the health economics associated with oral antiviral drugs in the treatment of CHB, and scant attention has been devoted to a comparative analysis of the cost–utility relationship between tenofovir alafenamide (TAF) and entecavir (ETV).” Can you reference these few studies?
-2. Materials and Methods
--2.3. Economic Evaluation: “The total direct medical costs refer to the one-year outpatient and inpatient costs of each patient after starting TAF of ETV.” Over what period of follow-up was the estimate of these costs obtained? If the use of TAF is very recent, the estimated costs may not be representative.
--2.4. Cost-utility Analysis & Figures 2-3: “The model classifies patients’ post-treatment (the first-line drug) conditions into four categories: CHB, hepatocellular carcinoma (HCC), cirrhosis, and death”. Also, Figures 2 and 3 show only these 4 health states. Does this mean there is no distinction between compensated and decompensated cirrhosis? However, health-related costs are very different depending on whether cirrhosis is compensated or decompensated. How is cirrhosis differentiated between compensated and decompensated in the model to be able to take into account the different utilities presented in Table 2?
--2.5. Statistical Analysis: An extended sensitivity analysis should be conducted and detailed. Univariate sensitivity analysis should be conducted on all model parameters. It would seem that a variation of +/-20% of the parameters was chosen but this is not what appears in Table 2 when we see the range of utility values. A multivariate sensitivity analysis including the most influential variables identified in the univariate analysis would be appreciated.
--Table 2: What does “fixed” mean for cost distribution? That they were not included in the PSA?
--Table 2 and Figure 8: Outpatient and hospital costs for each group (TAF and ETV) are defined according to the cost structure in Table 2 while they are expressed according to the state of health in Figure 8. How do we go from to one another?
--Page 8: “In addition to various costs, the required parameters of the Markov decision model in this study included the clinical transfer probability between states and the utility of each state. The value sources were all collected from the research and the probability, cost, and utility were compiled in Table 2 [14-18]. The constructed Markov decision model was presented in Figure 3. The optimum path selection for decision analysis was analyzed by using the Rollback method, which is a method used to backtrack the respective total costs and treatment results caused by choosing different strategies. The time horizon of the Markov decision model simulation was set to 30 years” This part should be moved to the Material and Methods section.
--Table 3 and related text: as written, the TAF group was less costly and more effective (higher QALY) when comparing to the ETV group; it is therefore not useful to calculate an ICUR: TAF strategy dominated ETV strategy and was cost saving.
--Figure 4: Why are there 3 points? One for TAF, one for ETV and the last one??? I'm not convinced by the usefulness of this Figure.
--Figures 4 to 6: Please keep the same representation: TAF is represented by a square (red) in Figure 4, and a triangle (red) in Figures 5 and 6, and the opposite in blue for ETV...
--Figure 7: The title should be “Incremental cost-effectiveness scatter plot of tenofovir alafenamide (TAF) versus entecavir (ETV) in the treatment of chronic hepatitis B (CHB) patients...”
--Figure 8:
---The title should specify which result is given: ICER of TAF compared to ETV?
---ICER or ICUR?
---Why are the variations centered around 0 and not around the EV?
---Why is EV equal to -NT$23,721/QALY and not -NT$23,878/QALY?
-Discussion
--page 14: “A synthesis of relevant studies is presented in Table 4 [23-26].” It is not usual to include a table in discussion and I am not convinced of its usefulness. Describing and discussing the important results seems sufficient to me.
--page 16: “ETV to TAF as the first-line drug could potentially save over NT$500 million annually (approximately US$18 million). This calculation is based on drug cost differences, the population estimate, and the proportion eligible for treatment ((NT$78,085 – NT$64,525) × 23,196,178 people × 0.9% × 19% = NT$537,863,697) [32] …”
Where do these drug costs come from? I can't find them in the manuscript... In addition, the calculation of a budgetary impact must include all costs related to the management of chronic hepatitis B, not just drug costs.
Author Response
REVIEWER 1
General comment:
The authors presented the cost-utility analysis of using TAF versus ETV in CHB patients based on real-world data, i.e. CHB patients treated with TAF or ETV (covered by the National Health Insurance), who visited a regional teaching hospital or a district teaching hospital in southern Taiwan from January 2016 through December 2020. Their main objective was to identify the optimal first-line oral antiviral drug in CHB patients. Data were collected retrospectively from medical records and health insurance claims: 54 patients in TAF group and 98 patients in ETV group. The analysis concludes that TAF is more cost-effective than ETV for treating CHB patients.
My general comment is that this conclusion was highly expected since progression is always faster and costs higher with ETV compared to TAF.
Ans: We have revised the statements and value your guidance on the Conclusions in our revised manuscript. Thank you.
My specific comments could help ensure the methodology applied and the robustness of the conclusion.
Specific comments:
-1. Introduction: “A limited number of studies have undertaken an exploration of the health economics associated with oral antiviral drugs in the treatment of CHB, and scant attention has been devoted to a comparative analysis of the cost–utility relationship between tenofovir alafenamide (TAF) and entecavir (ETV).” Can you reference these few studies?
Ans: At the end of the sentence above, references #6 to #8 have been cited. Thank you.
-2. Materials and Methods
--2.3. Economic Evaluation: “The total direct medical costs refer to the one-year outpatient and inpatient costs of each patient after starting TAF of ETV.” Over what period of follow-up was the estimate of these costs obtained? If the use of TAF is very recent, the estimated costs may not be representative.
Ans: In accordance with the provided advice above, we have incorporated the following statement into our revised manuscript: “The total direct medical costs denote the one-year outpatient and inpatient expenses per patient upon initiating either TAF or ETV treatment.” Thank you.
--2.4. Cost-utility Analysis & Figures 2-3: “The model classifies patients’ post-treatment (the first-line drug) conditions into four categories: CHB, hepatocellular carcinoma (HCC), cirrhosis, and death”. Also, Figures 2 and 3 show only these 4 health states. Does this mean there is no distinction between compensated and decompensated cirrhosis? However, health-related costs are very different depending on whether cirrhosis is compensated or decompensated. How is cirrhosis differentiated between compensated and decompensated in the model to be able to take into account the different utilities presented in Table 2?
Ans: The Markov model applied in this study was crafted through the integration of insights from local clinical practices and references to pertinent studies, as depicted in Figure 2. This model classifies the post-treatment conditions of patients receiving the first-line drug into four distinct states: chronic hepatitis B, hepatocellular carcinoma, cirrhosis, and mortality. The articulated Markov decision tree model is visually represented in Figure 3. The optimal path selection for decision analysis was scrutinized employing the Rollback method, a technique facilitating the retrospective tracing of total costs and treatment outcomes resulting from the implementation of distinct strategies. Notably, our analysis does not encompass a comparison of post-treatment conditions, including both compensated and decompensated cirrhosis, as well as associated health-related costs.
--2.5. Statistical Analysis: An extended sensitivity analysis should be conducted and detailed. Univariate sensitivity analysis should be conducted on all model parameters. It would seem that a variation of +/-20% of the parameters was chosen but this is not what appears in Table 2 when we see the range of utility values. A multivariate sensitivity analysis including the most influential variables identified in the univariate analysis would be appreciated.
Ans: In accordance with the provided advice above, we have incorporated the following statements into our revised manuscript: “The study devised a Markov decision tree model with predetermined parameters encompassing utility values, transfer rates, costs, and benefits for each state. Simulating patient state transitions during intervention and aggregating outcomes informed the identification of the more cost-effective alternative through Markov decision tree analysis. The probabilistic sensitivity analysis (PSA) comprised 1,000 Monte Carlo simulations, yielding ICUR values. Graphical representation employed cost–utility acceptability curves and scatter plots. A one-way sensitivity analysis, manipulating costs and out-comes by ±20%, discerned influential variables presented through tornado diagrams.” Again, thank you very much.
--Table 2: What does “fixed” mean for cost distribution? That they were not included in the PSA?
Ans: We apologize for the confusion. We have updated the cost distribution in Table 2 from “fixed” to “gamma” in Table 2. Thank you.
--Table 2 and Figure 8: Outpatient and hospital costs for each group (TAF and ETV) are defined according to the cost structure in Table 2 while they are expressed according to the state of health in Figure 8. How do we go from to one another?
Ans: Table 2 delineates the health insurance claims of patients one year after commencing antiviral drugs for hepatitis B, categorizing them into outpatient costs, inpatient costs, and total one-year medical costs. Conversely, Figure 8 presents a tornado diagram resulting from the one-way sensitivity analysis. Each parameter was scrutinized independently, and the outcomes indicated the order of significance for parameters influencing the incremental cost-utility ratio (ICUR). Again, thank you.
--Page 8: “In addition to various costs, the required parameters of the Markov decision model in this study included the clinical transfer probability between states and the utility of each state. The value sources were all collected from the research and the probability, cost, and utility were compiled in Table 2 [14-18]. The constructed Markov decision model was presented in Figure 3. The optimum path selection for decision analysis was analyzed by using the Rollback method, which is a method used to backtrack the respective total costs and treatment results caused by choosing different strategies. The time horizon of the Markov decision model simulation was set to 30 years” This part should be moved to the Material and Methods section.
Ans: Following your aforementioned advice above, we moved the statements into the Statistical Analysis section of our revised manuscript. Again, thank you.
--Table 3 and related text: as written, the TAF group was less costly and more effective (higher QALY) when comparing to the ETV group; it is therefore not useful to calculate an ICUR: TAF strategy dominated ETV strategy and was cost saving.
Ans: We put your words and rewrote the statements in our revised manuscript. Thank you.
--Figure 4: Why are there 3 points? One for TAF, one for ETV and the last one??? I'm not convinced by the usefulness of this Figure.
Ans: In alignment with the provided advice, Figure 4 has been removed from our revised manuscript. Thank you.
--Figures 4 to 6: Please keep the same representation: TAF is represented by a square (red) in Figure 4, and a triangle (red) in Figures 5 and 6, and the opposite in blue for ETV...
Ans: Figure 4 has been removed from our revised manuscript. Thank you.
--Figure 7: The title should be “Incremental cost-effectiveness scatter plot of tenofovir alafenamide (TAF) versus entecavir (ETV) in the treatment of chronic hepatitis B (CHB) patients...”
Ans: We put your words above in the title in Figure 7 (NEW Figure 6). Thank you.
--Figure 8:
---The title should specify which result is given: ICER of TAF compared to ETV?
---ICER or ICUR?
---Why are the variations centered around 0 and not around the EV?
---Why is EV equal to -NT$23,721/QALY and not -NT$23,878/QALY?
Ans: Firstly, we put your words into the title of Figure 8 (now NEW Figure 7). Secondly, we replaced the term "ICUR" with "ICER" throughout the entire content. Thirdly, in the tornado diagram, variations are centered around the expected values. We have revised Figure 8 (NEW Figure 7) accordingly in our revised manuscript. Finally, in the tornado diagram of Figure 8, the Expected Value (EV) of ICER closely aligns with the ICER value presented in Table 3. Again, thank you.
-Discussion
--page 14: “A synthesis of relevant studies is presented in Table 4 [23-26].” It is not usual to include a table in discussion and I am not convinced of its usefulness. Describing and discussing the important results seems sufficient to me.
Ans: In accordance with your advice above, Table 4 has been omitted from our revised manuscript. Thank you.
--page 16: “ETV to TAF as the first-line drug could potentially save over NT$500 million annually (approximately US$18 million). This calculation is based on drug cost differences, the population estimate, and the proportion eligible for treatment ((NT$78,085 – NT$64,525) × 23,196,178 people × 0.9% × 19% = NT$537,863,697) [32] …”
Where do these drug costs come from? I can't find them in the manuscript... In addition, the calculation of a budgetary impact must include all costs related to the management of chronic hepatitis B, not just drug costs.
Ans: The calculation relies on total direct medical costs, and the statements have been revised in our updated manuscript. Again, thank you.
Reviewer 2 Report
Comments and Suggestions for Authors
Hepatitis B is a huge concern for the world. The findings of the authors seems to impart a huge impact of the affordability of its treatment. However I believe there are some issues that must be answered by the authors:
1) Some of the drawback of use of interferon has been given by the authors. One drawback stated is the relapse of symptoms. However, it is not clear whether the use of TAF and ETV will reduce or eliminate the chances of relapse. The author should present some scientific proof.
2) The authors have used previously proven techniques to construct the proposed model such as EQ-5D-3L scores and Time Trade off (TTO). However, I feel just providing their references is not sufficient. The authors must briefly explain these techniques to elaborate how they obtained the scores using their data.
3) The authors have only used one Markov model to construct their results. To provide some comparative analysis other such non-parametric model may also be used. The authors have used the Beta distribution, it will be interesting to know that variation in results using other appropriate distributions such as Gamma.
4) The authors should incorporate some appropriate validation technique to substantiate their results.
5) Since the authors non parametric model therefore they must cite these papers which also make use of such models
"Khan, Yaser Daanial; Mahmood, M Khalid; Ahmad, Daud; Al-Zidi, Nasser M; ",Content-Based Image Retrieval Using Gamma Distribution and Mixture Model,Journal of Function Spaces,2022,,,2022,Hindawi
Comments on the Quality of English LanguageThe manuscript maybe reviewed for minor language errors.
Author Response
REVIEWER 2
Hepatitis B is a huge concern for the world. The findings of the authors seems to impart a huge impact of the affordability of its treatment. However I believe there are some issues that must be answered by the authors:
1) Some of the drawback of use of interferon has been given by the authors. One drawback stated is the relapse of symptoms. However, it is not clear whether the use of TAF and ETV will reduce or eliminate the chances of relapse. The author should present some scientific proof.
Ans: In response to the aforementioned advice above, we have incorporated and revised statements pertaining to the relapse of symptoms following the use of Tenofovir alafenamide (TAF) and Entecavir (ETV) in our revised manuscript. Thank you.
2) The authors have used previously proven techniques to construct the proposed model such as EQ-5D-3L scores and Time Trade off (TTO). However, I feel just providing their references is not sufficient. The authors must briefly explain these techniques to elaborate how they obtained the scores using their data.
Ans: In alignment with the aforementioned advice above, we have integrated pertinent statements regarding the construction of the proposed model, encompassing aspects such as EQ-5D-3L scores and the Time Trade-off (TTO) method, within the Statistical Analysis section of our revised manuscript. Again, thank you.
3) The authors have only used one Markov model to construct their results. To provide some comparative analysis other such non-parametric model may also be used. The authors have used the Beta distribution, it will be interesting to know that variation in results using other appropriate distributions such as Gamma.
Ans: We apologize for the confusion. We have updated the cost distribution in Table 2 from “fixed” to “gamma” in Table 2. Thank you.
4) The authors should incorporate some appropriate validation technique to substantiate their results.
Ans: Building on the provided advice above, the study developed a Markov decision tree model with predefined parameters for utility values, transfer rates, costs, and benefits in each state. Firstly, probabilistic sensitivity analysis (PSA) involved 1,000 Monte Carlo simulations, yielding Incremental Cost-Effectiveness Ratio (ICER) values. Graphical representation utilized cost–utility acceptability curves and scatter plots. Secondly, a one-way sensitivity analysis, varying costs and outcomes by ±20%, revealed influential variables with results presented through tornado diagrams. These statements have been added to the Statistical Analysis section of our revised manuscript. Thank you.
5) Since the authors non parametric model therefore they must cite these papers which also make use of such models
"Khan, Yaser Daanial; Mahmood, M Khalid; Ahmad, Daud; Al-Zidi, Nasser M; ",Content-Based Image Retrieval Using Gamma Distribution and Mixture Model, Journal of Function Spaces,2022,,,2022,Hindawi
Ans: Based on your advice above, we have included the pertinent references in our revised manuscript. Thank you.
Round 2
Reviewer 2 Report
Comments and Suggestions for Authors
The issues raised by me has been adequately answered. The paper maybe published.
Comments on the Quality of English LanguageThe English is good and understandable for the reader. However, I just that the manuscript might be thoroughly review for any minor language errors.